# CoRNeA: A Pipeline to Decrypt the Inter-Protein Interfaces from Amino Acid Sequence Information

**DOI:** 10.3390/biom10060938

**Published:** 2020-06-22

**Authors:** Kriti Chopra, Bhawna Burdak, Kaushal Sharma, Ajit Kembhavi, Shekhar C. Mande, Radha Chauhan

**Affiliations:** 1National Centre for Cell Science, Pune 411007, Maharashtra, India; krits04@gmail.com (K.C.); bhawna.burdak@gmail.com (B.B.); 2Inter-University Centre for Astronomy and Astrophysics, Pune 411007, Maharashtra, India; kaushalksharma1989@gmail.com (K.S.); akk@iucaa.in (A.K.); 3Council of Scientific and Industrial Research (CSIR), New Delhi 110001, India; shekhar.mande@gmail.com

**Keywords:** co-evolution, machine learning, inter-protein interfaces

## Abstract

Decrypting the interface residues of the protein complexes provides insight into the functions of the proteins and, hence, the overall cellular machinery. Computational methods have been devised in the past to predict the interface residues using amino acid sequence information, but all these methods have been majorly applied to predict for prokaryotic protein complexes. Since the composition and rate of evolution of the primary sequence is different between prokaryotes and eukaryotes, it is important to develop a method specifically for eukaryotic complexes. Here, we report a new hybrid pipeline for predicting the protein-protein interaction interfaces in a pairwise manner from the amino acid sequence information of the interacting proteins. It is based on the framework of Co-evolution, machine learning (Random Forest), and Network Analysis named CoRNeA trained specifically on eukaryotic protein complexes. We use Co-evolution, physicochemical properties, and contact potential as major group of features to train the Random Forest classifier. We also incorporate the intra-contact information of the individual proteins to eliminate false positives from the predictions keeping in mind that the amino acid sequence of a protein also holds information for its own folding and not only the interface propensities. Our prediction on example datasets shows that CoRNeA not only enhances the prediction of true interface residues but also reduces false positive rates significantly.

## 1. Introduction

The biological machinery performs its cellular functions when its basic units, such as DNA, RNA, and proteins, interact with each other. To understand the overall functioning of the cell, it is important to delineate the pairwise interactions of these basic units such as DNA-protein, RNA-protein, and protein-protein. Of these, the inter-protein interactions that a cell possesses play a very crucial role in understanding the various cellular processes and hence also their functioning or misfunctioning in the disease models. There are various experimental methods known for examining these interactions such as yeast two hybrid (Y2H) [1], co-immunoprecipitation (co-IP) [2], mass spectrometry [3], etc., which provide information only about the domains necessary for maintaining the interaction or the proximity of the interactions. Moreover, these methods are labor, cost and time intensive. Deciphering the PPII (Protein-Protein Interaction Interfaces) at the highest resolution through x-ray crystallography or cryo-electron microscopy methods is even more challenging due to their intrinsic technical difficulties.

A number of in silico methods have been described earlier to predict these PPII based on available data, such as (1) homology-, (2) machine learning-, and (3) Co-evolution-based. Homology based methods are generally applied when confident homologs of both the interacting proteins are available, followed by protein-protein docking for visualizing the protein interaction interfaces, such as PredUS [4], PS-HomPPI [5], PriSE [6], etc. The machine learning (ML) methods which have been described till date are either structure-based or sequence-based. The structure-based ML methods (such as SPPIDER [7], PINUP [8], PAIRpred [9], PIER [10], ProMate [11], Cons-PPISP [12], Meta-PPISP [13], CPORT [14], WHISCY [15], InterProSurf [16], VORFFIP [17], eFindSite [18], etc.) require three-dimensional information of the interacting proteins which can be either experimental or homology driven to incorporate the geometrical complementarities of amino acids as training features. Only a few sequence-based ML methods are known, such as BIPSPI [19], PSIVER [20], and ComplexContact [21], which derive features based on conservation, physicochemical properties of amino acids, etc. However, the predictability of these ML methods is affected by the prevalence of high false-positive rates due to limitation of small number of protein-complex structures in the protein structure database (PDB) which restrict the training of these machine learning algorithms in terms of variability.

The third class, Co-evolution-based methods which were originally formulated to predict contact forming residues within a single protein and therefore for the prediction of the structure of the protein. These methods have been extrapolated to also predict the inter-protein interaction interfaces based on the multiple sequence alignments (MSA) of the proteins. Concatenating the MSA of an interacting pair and using the same statistical formulae as described for intra pairs have been implemented to predict the co-evolving contact forming pairs by various methods, such as DCA [22], EvComplex [23], etc. However, there are two main caveats known for these methods. Firstly, they use different downstream methods to filter out their results by using homology-based models and docking predictions in combination with their results. Secondly, most of these methods have been tested on prokaryotic proteins and have a limitation of predicting only for a maximum combined length of 1500 residues per protein pair. Almost all Co-evolution-based methods have been only tested on prokaryotic lineage probably due to availability of huge number of sequences for generating variable multiple sequence alignments. Recently, a hybrid method (Co-evolution and machine learning based- ComplexContact [21]) was reported; however, its performance was also tested on prokaryotic datasets. Additionally, maximum methods developed till date which utilize either sequence or structural information as input and depict their predictions in terms of propensity of a residue to be present in the interface as single residue scores. There are only a few exceptions present in all classes of methods described above, such as PS-HomPPI [5] for homology-based methods, PAIRpred [9], BIPSPI [19], and ComplexContact [21] for machine learning methods and all the Co-evolution based methods (Table 1), which depict their predictions in terms of pairwise scores. A comparison of all the machine learning methods suggests that predictions are more specific if presented in a pairwise manner (such as BIPSPI [19] and PAIRpred [9]).

Although the low predictability of the methods available to date for eukaryotic protein complexes can be attributed to the differences in the rate of evolution of the proteins in the two lineages. It has been reported that there is a difference in the composition of the type of amino acids present in prokaryotic versus eukaryotic proteins and also in the radius of gyration and planarity in the interaction interface. Since the eukaryotic proteins are not exclusive to only one set of function, it has been perceived that most of the eukaryotic protein interactions are transient, having smaller interaction hotspot zones and more planar binding sites consisting of more polar and aromatic residues. These properties of the eukaryotic protein interactions make them essential part of cell signaling pathways [24].

Hence, to delineate the vast PPII network of eukaryote lineage, e.g., human protein interaction network, which contains about one hundred and fifty thousand (150,000) interactions (with only about 10% of known structures of these protein complexes) [25], it is important to develop a method specific for eukaryotic predictions. In this report, we present a new hybrid pipeline based on the framework of Co-evolution, Random Forest (ML method), and Network Analysis (CoRNeA) for predicting the pairwise residues of the PPII from the protein sequence information of two interacting proteins (Figure 1). We also developed a new hybrid method for calculating co-evolving positions in the interacting pairs based on mutual information and Statistical Coupling Analysis (SCA) [26]. Owing to high signal to noise ratio, this method in consensus with the other Co-evolution-based method does not perform well independently to extract the precise interacting pair of residues especially for eukaryotic proteins. Hence, we used this method as one of the features for machine learning pipeline. The other features derived for the Random Forest classifier are based on the physicochemical properties of the amino acids which depend on their side chain structure, such as charge, size and hydrophobe compatibility, secondary structure information, and relative solvent accessibility, were also derived using amino acid sequence information. To include the energetics of interactions, contact potentials were also included as features. Similar to other machine learning classifiers, our pipeline also predicted a number of false positives. In order to reduce them, we employed network analysis by incorporating the intra-contact information to generate residual networks for PPII. In summary, the major highlight of this method as compared to other methods developed on the similar lines are: (1) use of eukaryotic protein structure database for training the classifier; (2) use of Co-evolution information as evolution-based feature; and (3) use of intra-contact pairs to eliminate false positive pairs through network analysis. Thus, we present a holistic approach to this complex problem of identifying pair of residues forming the interaction interface in the heterodimers from the amino acid sequence information.

## 2. Methodology

The overall pipeline to predict pairwise contact forming residues from sequence derived data can be divided into three distinct parts as depicted in Figure 1. The first step is to generate pairwise features (Co-evolution, structural and contact potential based) from the amino acid sequence of the two interacting proteins (Figure 1A–C). The second step is to feed these pairwise features in a Random Forest classifier, hence optimizing its various hyperparameters to obtain the best evaluation statistics (Figure 1D). The third step is to combine the intra-protein contact forming residues from Co-evolution-based method and inter-protein contact forming residues from Random Forest classifier and perform network analysis to predict the exclusive pair of residues forming the interface of the two interacting proteins (Figure 1E–G).

### 2.1. Datasets

The Affinity Database version 2.0 [27] was used to select the protein complex structures for training (42 complexes were selected for training). The amino acid sequences of the complex structures were extracted from [28] and used as a query to search for homologs. PHMMER [29] was used to fetch maximum homologs of the query sequence, which were then manually curated to remove redundant sequences. The sequences having less than 25% sequence identity were removed. The final dataset for each of the interacting protein consisted of identical species.

### 2.2. Multiple Sequence Alignments

The datasets for each interacting pair of proteins having identical species were subjected to structure-guided multiple sequence alignments using PROMALS3D [30]. The alignments were then analyzed/edited in JalView version 2.11.0 [31] and then concatenated (last residue of Protein A followed by first residue of Protein B) in R version 3.6.1 using package seqinr [32]. These concatenated MSA datasets were used for Co-evolution matrix calculations.

### 2.3. Features

For calculating sequence-based features, the sequences were extracted from the protein databank (www.rcsb.org) and any missing regions reported in the structure were removed from the sequence data. All the features for training and testing were compiled as all versus all residue pairs between sequence of the interacting pair of protein (Protein A and Protein B) in form of M × N matrix (*M* = length of Protein A and *N* = length of Protein B). All the feature values were scaled between 0 and 1 (Appendix A).

#### 2.3.1. Evolution Based Features

##### Co-Evolution Matrices (CMI)

The Co-evolution scores between the pair of residues of the interacting proteins were calculated based on Conditional Mutual Information as depicted in Figure 2. The concatenated MSA’s were subjected to perturbation experiment similar to that used in Statistical Coupling Analysis (SCA) [26]. The amino acids were converted from alphabetic nomenclature to numeric for the ease of calculation (Appendix A). For each column in the MSA of Protein A and B, a condition pertaining to the presence of one of the 20 amino acid was given to subset the concatenated MSA. For example, position 1 in concatenated MSA, a condition given to subset the MSA for the presence of valine (V). A subset of sequences was selected which had only valine at position 1 of MSA. Frequencies of the amino acid present in the subset were calculated and subjected to the conditional mutual information formula [33]. It resulted in 20 such conditions for each column in the MSA of Protein A, which were summed up to obtain the final Co-evolution M × N matrix.

#### 2.3.2. Structure Based Features

##### Charge, Hydrophobe, and Size Compatibility Matrices

The physicochemical properties of the residue determined by the composition and chemical structure were used to derive the structure-based features. These features can be derived from sequence information but to derive pair wise values for these properties, we employed the 20 × 20 residue matrices which were described to aid in ab initio modeling of single protein [34]. These matrices were used to derive an all versus all residue matrix (M × N) for the interacting pair of proteins as features, i.e., hydropathy compatibility (HCM), charge compatibility (CCM), and size compatibility matrices (SCM).

##### Relative Solvent Accessibility (RSA)

To calculate the pairwise RSA values, RSA of independent proteins were calculated using SPIDER3 [35] and multiplied to form an all versus all (M × N) matrix of the pair of interacting proteins.

##### Secondary Structure Predictions (SSP)

The secondary structure of the proteins was predicted using PSIPRED version 3.3 [36] and all residues were assigned numbers (i.e., 1 = α-helix, 2 = β-sheet, and 3 = l-loop). Simple multiplication and scaling of these numbers between 0 and 1 would yield in a combination where α-helix to α-helix instance will be ranked lowest. To avoid this mis scaling, the training dataset was inspected for the nature of residue-residue combinations in terms of secondary structures and the 6 possible combinations (i.e., α-α, α-β, α-l, β-β, β-l, and l-l) were ranked in order of occurrence. These values were then used as standard to fill in all M × N matrices of the two interacting proteins.

#### 2.3.3. Contact Potential Based Features

Three different approximations of contact potentials were used to generate contact potential-based features. The first approximation was the original matrix (MJ matrix) [37] where the effective inter-residue contact energies for all amino acid pairs were calculated based on the statistical analysis of protein structures. The other two approximations were derived from the MJ matrix, where a 2-body correction was applied on this matrix to generate two separate matrices [38]. One of them was specific for capturing the interactions between exposed residues and the other one for buried residues. Thus, all three possible combinations were used to derive three contact potential (M × N) matrices, namely CP: original MJ matrix, CPE: MJ matrix derived for exposed residues, and CPB: MJ matric derived for buried residues, for the pair of interacting proteins.

### 2.4. Environment Features

To include residue environment information for training the machine learning algorithm, a kernel matrix of size 5 × 5 was defined and convolved over the nine feature matrices as described above. The convoluted features were generated by using OpenImageR (https://github.com/mlampros/OpenImageR) package in R and the size of the matrices were kept same to avoid any loss of information. Additionally, various other kernel matrices were also used to train and test different datasets varying from 3 × 3 to 7 × 7 with varying percentage decrease in the weights from 10% to 25%. Hence, for each independent training/testing cycle, 18 feature matrices were used for each pair of interacting protein for training the Random Forest classifier (9 original features and 9 derived features).

### 2.5. Interface Residue Labeling

The interface residues for the protein complexes were extracted using PISA version 1.52 [39]. The number of residue pairs present in the interface (five hundred pairs for 42 complexes) was far less than all possible residue pairs of the two interacting proteins (two million pairs for 42 complexes). To increase the search space and take into consideration the environment of the contact forming residues, a distance cut off of 10 Å was used to search for possible pair of residues flanking −2 to +2 positions of the interface residues extracted from PISA. This yielded ten times more positive labels (5000 pairs for 42 complexes) for training the classifier.

### 2.6. Data Imbalance Problem

Although increasing the search space as explained above yielded 10 times more data points, still the complete protein complex database exhibited highly imbalanced data. Five thousand pairs were labeled as positive out of the total two million pairs. In order to address this imbalance class problem, the majority class, which was the negative data labels (non-interface residues pairs) was down sampled. A number of ratios for negative to positive samples were tested iteratively (e.g., 2:1, 5:1, 10:1, and 20:1) and best evaluation statistics were obtained when the negative sample size was five times that of positive samples (5:1). This was used as training set for the supervised classification model.

### 2.7. Random Forest Classifier

The Random Forest classifier [40] was trained first using a grid search to optimize the hyperparameters for the model yielding the best evaluation statistics through cross-validation. The hyperparameters obtained from the grid search were then used to train the classifier with training to test sample split to 75:25. The scoring function used for optimizing the hyperparameters was chosen as F1 score owing to imbalanced nature of the dataset used for training. Scikit-learn [41] was used to import the Random Forest classifier base algorithm. Training was performed on the same datasets both with and without environment features. All the datasets were compiled using R and Rstudio (http://www.rstudio.com/) and machine learning was performed using Python version 3.7 via anaconda-navigator (https://anaconda.com).

### 2.8. Performance Evaluation

To determine the predictability of the classifier models, leave-one-out method of cross validation was applied were the training dataset consisted of sampled pairs from 41 complexes and the 42nd protein complex was treated as test sample. To maintain homogeneity for comparing the results of all 42 protein complexes, the 5 × 5 feature kernel matrix was used to derive the environmental features. Owing to the imbalance nature of the number of interacting pairs versus non interacting pairs for any given protein complex used for testing, the evaluation parameters used for determining the predictability of the model was both macro and weighted averages of precision, recall and F1-score. The macro averages for each of these parameters gives equal importance to both the data classes despite the imbalance nature of the dataset, whereas the weighted averages considers class imbalance while calculating the evaluation parameters.

### 2.9. Network Analysis

To reduce the number of false positives obtained from the Random Forest classifier, a holistic approach was adopted as described in Figure 3 to include the intra-protein predictions. To determine the intra-contacts, we used the Co-evolution method as described in Section 2.3.1 by concatenating Protein A with itself (similarly for Protein B) (Figure 3B). To determine the contact forming intra-protein residue pairs, the residues present at a sequential distance less than 5 residues were eliminated and only top 5% of the coevolution values were taken as positive. The residue pairs obtained from this analysis for both proteins were used to plot the intra-protein residue networks in Rstudio using igraph package [42].

The predictions from the Random Forest classifier were used to plot the inter-protein residue network as a bipartite graph using the igraph package in Rstudio. Since the RSA for residues present in the core of the protein should be 0, these residues were extracted from SPIDER3 [35] for both the proteins independently. A residual network was hence computed for the inter-protein contact predictions by first eliminating the nodes representing RSA = 0 and then the intra-protein contacts from Protein A and B (Figure 3C,D).

### 2.10. Scoring of Positive Pairs Using Convolution Feature Matrix

The residual inter-protein network obtained were then plotted as a binary matrix of Protein A versus Protein B where 0 represented predicted non interface pairs and 1 represented predicted interface pairs. To identify the most probable interaction interfaces, cluster of 1′s was identified by convolving a unitary matrix of size equal to that of kernel matrix used for deriving environmental features (i.e., 3 × 3 or 5 × 5) over the prediction matrix. Subsections having the maximum number of 1′s hence obtained the highest score (score of 9 for 3 × 3 matrix and 25 for 5 × 5 matrix). A cut off value of 2 for 3 × 3 matrix and 6 for 5 × 5 matrix was selected to sort the high scoring pairs considering that at least 25% of the 3 × 3 or 5 × 5 subsections of the prediction matrix are populated with 1′s. These high scoring pairs were then extracted and mapped onto the test dataset structures to identify the true positives such that they also occur in the group of 3 residues at a stretch in both the proteins.

### 2.11. Immunoprecipitation for Validating Interface Residues

Human Nup93 (KIAA0095) fragments (full length (1–819), 1–150, 1–82, 96–150) were cloned in pEGFP-C1 expression vector (Clontech, Mountain View, CA, USA) fused with GFP at N-terminus. HEK293F cells (Invitrogen, Carlsbad, CA, USA) cultured in freestyle media (Gibco, Grand island, NY, USA) in a humidified incubator maintained with 8% CO_2_, 37 °C at 110 rpm, were transfected with plasmid DNA using Polyethylenimine (Polysciences, Warrington, PA, USA). Cells were harvested after 60 h and lysed with lysis buffer (1X DPBS (Gibco), 0.2% tween 20, protease inhibitor cocktail, 1mM PMSF) by incubating the cells on ice for 30 min followed sonication and centrifugation. 1 mg of supernatant was incubated with glutathione beads (Pierce, Rockford, IL, USA) pre-bound with GST tagged Anti-GFP nanobody (Addgene ID # 61838, Watertown, MA, USA) [43] for 4 h and 5% lysate was taken as input. The beads were then washed with lysis buffer thrice and the pulled fractions were eluted by incubating with elution buffer (1X DPBS, 50 mM Tris Cl pH 8, 150 mM NaCl, 0.5 mM EDTA, 5 mM β-mercaptoethanol, 10 mM reduced glutathione. Eluted fractions were separated on 10% SDS PAGE, and transferred onto PVDF membrane (Millipore, County Cork, Ireland). Blots were then probed with primary antibody Anti-Nup205 at 1:4000 (Sigma HPA024574, Saint Louis, MO, USA), Anti-GFP 1:3000 (Sigma G1546) followed by secondary HRP conjugate. Blots were developed using Quant HRP substrate (Takara, Kusatsu, Japan) and images were acquired on Amersham Imager 600 (GE).

## 3. Result and Discussion

### 3.1. Feature Derivation

The predictability of any supervised machine learning method is dependent on the nature of features used for training. Random Forest classifier is a tree-structure based algorithm where the classification rules are learned based on the feature values and their target class provided while training. Various features generated for training the Random Forest classifier were divided into three categories viz evolution-based, structure-based (physicochemical properties derived from structure of amino acid side chains) and contact potential-based features. For the evolution-based feature, a new Co-evolution algorithm was derived as explained in Section 2.3.1 and Figure 2. The new method as described in Section 2.3.1 provided better scores for the interface residues as opposed to other Co-evolution methods (Appendix A). Another important difference was generation of only a single non-symmetric M × N matrix from this method as opposed to L × L (where L = M + N) from other methods which result in higher signal to noise ratios. Thus, the conditional mutual information (CMI) based method was able to provide more confidence to the co-evolving pair of residues and decreasing the noise by generating the M × N matrices. Moreover, the co-evolving pair of residues in the interacting proteins maintains the homeostasis of the interaction across species, hence using them as a feature, as opposed to the standard PSSM-based conservation methods (such as PAIRpred [9], eFindSite [18], Cons-PPISP [12], PSIVER [20], BIPSPI [19], etc.) provided better predictability.

The nature of physicochemical properties of the residue interaction in the protein interface is somewhere in between their properties when present in the core or on the surface of the protein. It has been reported that the interface environment is closer to that exhibited on the outside in contact with the solvent as opposed to that present in the core of the protein [44]. For example, relative solvent accessibility of a residue which defines its possible position in the protein, i.e., whether it will be present in the core of the protein (relative solvent accessibility of 0) or is solvent-exposed (relative solvent accessibility >0). For the residues which lie in the PPI interface, they should have value as 0 < RSA < 1 if the value is scaled between 0 and 1. Due to lack of specific standard matrices for inter-protein residue contacts, those derived for intra-protein contacts were used for feature generation in this method, which includes charge, hydrophobe and size compatibilities, relative solvent accessibility, and secondary structure predictions.

The knowledge-based statistical potentials have also been used previously to mimic the interactions between the amino acids in a protein. One of such knowledge-based potential is the contact potential derived by Miyazawa and Jernigan based on statistical analysis of the protein structures. These contact potentials are widely used in the computational prediction for protein folding. The contact potentials for the residue lying in the PPI interface should ideally lie in between those of buried and exposed residues. To assess their applicability in identifying interface residues of the interacting proteins three approximations of these contact potentials were used as features.

The contacts between two residues of the interacting proteins also depend on its neighboring residues by creating a favorable niche for the interaction to take place. Hence, the properties governing the interaction (as described above) of the neighboring residues will also have an impact on the overall predictability of the Random Forest classifier. To address this, the Random Forest classifier was trained in two different modes, i.e., with and without environment features, the results of which are explained below.

### 3.2. Evaluation of Environment Features in Random Forest Classifier

To validate the effect of the environment features on the Random Forest classifier, the classifier was trained both with and without the environment features. The overall accuracy obtained for the dataset trained with the environment features was 86% as opposed to that for without environment features was 80%. In terms of evaluation statistics, such as precision and recall, there an increase of 5% when environmental features are used to train the model (precision: 86% and recall 85%) as opposed to the one without environmental features (precision: 81% and recall 81%) as depicted in Table 2. The Receiver-Operator Curve and confusion matrix for five-fold cross-validation for the dataset without environment features (Appendix A) and with environment (Appendix A), as well as the other evaluation statistics, depicts that the Random Forest classifier predicts with better precision and recall, and hence F1 measure, when the environment features are used for training, thus validating that these derived features (environment features) are important in predicting the contact forming residue pairs for the interacting proteins.

### 3.3. Feature Importance Evaluation

One of the marked features of Random Forest classifier is that it is able to decipher the importance of every feature used for training, which can be used to determine the over-fitting of a model, as well as to gain insights about the physical relevance of the features in predicting the PPI interface. The feature importance plot for the dataset without the environment features (Appendix A) depicts that the three most important features are relative solvent accessibility (RSA), Co-evolution scores (CMI), and the contact potentials (CP). However, the feature importance plot for the dataset with environment features (18 features in all) (Figure 4), depicts the importance of these derived features. Of the 18 features used for training, the top 12 positions have all 9 derived/environment features, along with RSA, CMI, and CP. Thus, it is evident that all these features play a crucial role in the prediction of protein interaction interfaces.

### 3.4. Relationship between the Size of Feature Kernel Matrix and Type of Secondary Structures in the Interaction Hotspots

The interaction interfaces of the proteins can be classified into 6 possible categories based on the secondary structure compositions of the interface hotspot regions, such as α-α, α-β, α-l, β-β, β-l, and l-l (where α denotes helices, β denoted sheets, and l denoted loops). Since the residue environment features were identified as the most critical features in the training of Random Forest classifier model, it is important to consider the role of the size of kernel matrix used for training the classifier. The residue environment for any protein can range from n − 1 to n + 1 position and up to n − 3 to n + 3 positions; thus, all such variations were tested by training different classifiers. For every different size and weight of the feature kernel matrix, the derived features were generated and used to train different Random Forest models. For each of the test dataset, all these different models were tested to determine a relationship between the nature of interaction in terms of secondary structure pairs and the size and weight of feature kernel matrices. The optimized models were then utilized to test for pair of interacting proteins with known crystal structure which were not a part of the training dataset to validate the predictability of the method. As observed from Appendix A, for interface hotspots consisting of loop-loop or loop-sheet, interactions were predicted better using 5 × 5 kernel matrix derived model, and those consisting of helix-helix interfaces were predicted better using the 3 × 3 kernel matrix derived model.

### 3.5. Evaluation of Random Forest Trained Models

Leave-one out method as explained in Section 2.8 was used to test the machine learning component of the pipeline CoRNeA (i.e., Random Forest classifier). The performance evaluation of macro and weighted averages of precision, recall and f1-score, as well as accuracy, suggest that the features used for the training of the classifier models are able to predict the true positives, i.e., higher recall macro averages but the false positive discovery rate is higher, i.e., lower precision macro averages are observed (Appendix A). Additionally, if the test dataset is the part of the trained model, i.e., the sampled pairs from all 42 complexes (positive to negative sample ratio of 1:5) are used for generating the classifier model and all the 42 complexes are used as independent test datasets, the recall macro average values are increased at least two-fold, but the precision macro averages are still the same indicating higher false positive prediction (Appendix A). Hence, it can be concluded that, if a particular dataset has some sequence similarity with the training dataset complexes, both the true positive and false positive prediction rate will be higher. Hence, downstream reduction in the false positive rate by utilizing network analysis component of CoRNeA is of extreme importance. Overall, the machine learning component of CoRNeA can predict with an average accuracy of 89% on the test datasets, which can be further improved by using the network analysis component of this pipeline.

### 3.6. Application/Evaluation of CoRNeA on Test Datasets with Known 3D Structures

Four eukaryotic protein complexes with known 3D crystal structures were used to assess the predictability of the complete pipeline CoRNeA. The interface residues of these protein complexes were predicted using model trained on 42 protein complexes of the affinity database. The combined amino acid length of the two proteins in these hetero dimers ranged from 127 amino acids to 986 amino acids. Additionally, owing to the variability in terms of secondary structure composition of these complexes, the derived features were generated using different sizes and weights of the kernel matrices for each test case. The model which predicted with the best evaluation statistics for each test case was considered for the downstream network analysis and final prediction matrix processing. Moreover, CoRNeA was used to predict the interaction interface of a known interacting pair of protein from the nuclear pore complex to assess the applicability of the pipeline to reduce false positive pairs in absence of structural information.

#### 3.6.1. Vav and Grb2 Sh3 Domain Heterodimer (PDB ID: 1GCQ)

The first test dataset used for assessing the predictability of CoRNeA was the crystal structure of Vav and Grb2 Sh3 domain (PDB ID: 1GCQ) [45] which consists of three chains. One of Vav proto-oncogene (Chain C) and the other two of growth factor receptor-bound protein 2 (Chain A and Chain B). The dataset was compiled for this protein pair using Chain A and Chain C of 1GCQ as query. The features were calculated as described above and used as test dataset for evaluating the trained Random Forest models with environment features. The total size of the dataset created by these two chains amounted to 4002 pairs of residues. The Random Forest classifier predicted 25 pairs correctly as true positives and 967 pairs were predicted as false positives.

To further reduce the number of false-positive pairs, network analysis was performed. The intra-protein contact forming residue pairs for Chain A (Protein A) and Chain C (Protein B) of 1GCQ were obtained from Co-evolution analysis where only top 5% pairwise values were considered to be true cases. The length of Chain A is 56 amino acids which would lead to 1540 intra pairs. The highest scoring 157 pairs were considered while constructing the intra-protein contact forming residue network of Chain A of 1GCQ, as depicted in Appendix A. The length of Chain C is 69 amino acids which would lead to 2346 intra-protein pairs. The highest scoring 238 pairs were considered while constructing the intra-protein contact forming network of Chain C of 1GCQ as depicted in Appendix A. The inter-protein contact forming residue pair network of Chain A and Chain C as obtained from Random Forest classifier is shown in Appendix A which consisted to 992 predicted pairs of which 967 were false positives. A residual network was calculated from the three networks mentioned above (as shown in Appendix A) and the final pairs were plotted as a matrix of Protein A versus Protein B. Since a 5 × 5 matrix was used to derive the environmental features, a unitary matrix of 5 × 5 was convolved onto the resultant interface prediction matrix. Pairs having convolved value more than 6 were selected, which reduced the total pairs to 359 of which 42 were true positives and 317 were false positives. The results obtained from the pipeline are shown onto the structure of Vav and Grb2 Sh3 domains (PDB ID 1GCQ) (Figure 5A(i)). Interestingly, the data labels provided while testing was only for Chain A and Chain C but the labels obtained after prediction were for both the pairs, i.e., Chain A and Chain C, as well as Chain B and Chain C (Appendix A), within 10 Å distance. In comparison to the interface predicted by PISA using the structural information, CoRNeA was able to predict at least 50% of true pairs as depicted in Figure 5A(ii). Thus, the overall pipeline to predict the PPI interface is fair in predicting the probable pairs of interacting residues, as well as to separate out the residue which might reside on the surface of the protein from those present in the core of the individual proteins only from amino acid sequence information. The confusion matrix before and after the network analysis for all the test datasets is provided in Table 3.

#### 3.6.2. Alpha Gamma Heterodimer of Human Isocitrate Dehydrogenase (IDH3) (PDB ID: 5YVT)

The structure of the alpha gamma heterodimer of human IDH3 (PDB ID: 5YVT) [46] (Figure 5B) was used as a second test dataset. This protein complex is from mitochondrial origin and its length (M + N) is larger (693 amino acids) as compared to the previous example (PDB ID: 1GCQ, 127 amino acids). Network analysis was performed for this dataset by calculating the intra-contacts of both chains A and B. The residual network resulted in 992 edges which were then mapped back in the form of the matrix of Protein A versus Protein B. A unitary matrix of 5 × 5 was convolved onto the predicted matrix and 537 pairs having value more than 6 were selected for analysis. Of these, 30 pairs formed the actual contacts when mapped onto the structure having distance within 10 Å, as shown in Figure 5B. Hence, this new pipeline can be used for proteins from eukaryotic origin, and the length of the pair of proteins in consideration is not a limiting factor.

#### 3.6.3. Ubiquitin Like Activating Enzyme E1A and E1B (PDB ID: 1Y8R)

The crystal structure of ubiquitin-like activating enzyme E1A and E1B (PDB ID: 1Y8R [47]) having a combined length of 986 amino acids (Protein A: 346 amino acids and Protein B: 640 amino acids) was used as another test dataset. Network analysis was performed for this dataset by calculating the intra-contacts of both chains A and B. The residual network resulted in 1166 edges which were then mapped back in the form of the matrix of Protein A versus Protein B. A unitary matrix of 3 × 3 was convolved onto the predicted matrix owing to the occurrence of α helical structure of the pair of proteins under consideration, resulting in total number of 898 positives pairs, of which 18 were true positives, and the remaining 880 were false positives (Figure 5C).

#### 3.6.4. Nup107-Nup133 Heterodimer of the Outer Ring of the Nuclear Pore Complex (PDB ID: 3CQC)

The crystal structure of Nup107-Nup133 complex (Nup107: 270 amino acids, Nup133: 227 amino acids, combined length of 497 amino acids) consists of the C-terminal region of both the proteins was used as another test dataset. The residual network consisting of 540 pairs was generated after removing the nodes which are a part of the intra network in either of the proteins. The total number of points were further reduced to 240 after performing convolution on the final prediction matrix using a unitary 3 × 3 matrix and keeping a cut off of more than 2. Of the 240 pairs, 6 pairs were identified as true positives within the distance of 10 Å (Figure 5D).

#### 3.6.5. Nup93-Nup205 Complex of the Adapter Ring of the Nuclear Pore Complex (NPC)

To test the applicability of the pipeline on the dataset without known structural information, hNup93-hNup205 interaction interface was explored. Nup93 is a linker protein of the Nup93-subcomplex of the NPC. It is known to connect the adaptor/ inner ring of the spoke region with the central channel pore of the NPC [48]. The adaptor region consists of the four proteins viz., Nup188, Nup205, Nup35, and Nup155. In terms of the known interactions of the specific domains of the Nup93, its R1 region which spans the first 82 amino acids is known to interact with the Nup62 of the central channel [49]. Nup93 is specifically known to form mutually exclusive complexes with either Nup188 or Nup205 of the adapter ring [49,50]. The interaction interface information for these pair of proteins is not known specifically from mammalian origin owing to difficulties in biochemical reconstitution of these complexes. However, for hNup93-hNup205, proximity information for this pair of proteins is known through crosslinking based mass spectrometry analysis [51]. The cross-linking data suggests three different regions of Nup93 to be in proximity of Nup205 (i.e., N-terminal, middle and C-terminal), but the most prominent hits are seen between the R2 (96–150) region at the N-terminal of Nup93 with the C-terminal of Nup205 (Figure 6A).

CoRNeA was employed to identify the interaction interface of Nup93-Nup205 complex by utilizing full length sequence information of both the proteins (Nup93: 819 amino acids and Nup205: 2012 amino acids). Since, the secondary structure prediction of both these proteins depicts α-helices, the 3 × 3 kernel matrix derived Random Forest model was utilized to predict the interface pairs. The resultant high scoring pairs, which pertained to specifically the R2 region of Nup93 (96–150) with the C-terminal region of Nup205 obtained from CoRNeA (Figure 6B), are in consensus with cross-linking mass spectrometry analysis (Appendix A). However various low scoring pairs were also identified for Nup93 middle and C-terminal region, but they did not span more than three continuous pairs (such as 89–91 of Nup93 with 1201–1205 of Nup205) between the two proteins.

Further, validation of the interacting interface between Nup93 and Nup205 predicted with CoRNeA analysis was done by in vitro pull-down experiment using Nup93 deletion constructs (Figure 6C). Upon pull down with GST tagged anti-GFP nanobody, the N-terminal region of Nup93 (1–150) was able to pull endogenous Nup205 efficiently. Further mapping the minimal interaction region, R2 fragment of Nup93 (96–150) was found to interact with endogenous Nup205, thus validating the in silico prediction by CoRNeA. A diminished interaction of the Nup93 region (176–819) was also observed through this pull-down experiment (Figure 6D), which is also consistent with the identification of low scoring regions identified by CoRNeA. This experimental validation depicts that CoRNeA is able to predict the short stretches of interaction hotspots between known pair of interacting proteins from only their sequence information and hence can be used to decipher the minimal interacting regions of pair of large proteins, thus aiding in their biochemical reconstitution followed by structural elucidation.

### 3.7. Comparison with Other Methods

To assess the predictability of CoRNeA, the results obtained from it for the four test cases described above were compared to the predictions of recently published method BIPSPI [19] and ComplexContact [21] which are closest to our implementation and the only available servers to predict the interface residues using only amino acid sequence information. BIPSPI utilizes physiochemical properties, as well as residue environment information, through hot encodings. Although the major point of difference between BIPSPI and CoRNeA lies in the choice of evolution-based feature (PSSM in BIPSPI versus Co-evolution in CoRNeA) and derivation of the environmental features (hot encoding in BIPSPI versus convolution averaging in CoRNeA). On the other hand, ComplexContact is based on the framework of Co-evolution and deep learning. It utilizes mutual information formulae for including Co-evolution information from concatenated MSAs. The deep learning model, however, is trained for intra-contact predictions including features, such as secondary structures and relative solvent accessibility, which are extrapolated to predict inter residue contact by including Co-evolution as an additional feature. However, the network analysis post processing of the results to remove the intra-contacts is one of the unique attributes of the pipeline CoRNeA which is not present with other machine learning based methods known for predicting the interaction interfaces. Since CoRNeA utilizes only the amino acid sequence information, the sequence mode of prediction on BIPSPI and ComplexContact servers were employed for predicting the interface residues of the four test datasets (PDB ID: 1GCQ, 5YVT, 1Y8R, and 3CQC). The Nup93-Nup205 dataset could not be processed using either BIPSPI or ComplexContact owing to its limitation to consider proteins larger than 1500 amino acids in length in case of BIPSPI and 1000 residues in case of ComplexContact. The results obtained for these datasets depicted that the final predictions from CoRNeA yielded in fewer false positives than BIPSPI, hence validating the overall improvement in the accuracy of the prediction of PPI interface residues. Interestingly, ComplexContact predicted very few pairs with probability of 0.5 and above for these test datasets but none of them were a part of the interface hotspots for the three test cases (PDB ID: 1GCQ, 5YVT, and 3CQC) and only one true pair for test set 1Y8R (Table 4).

Since comparable predictions in terms of number of true positives were only obtained for one test dataset (PDB ID: 5YVT), the results obtained from both CoRNeA and BIPSPI for this protein complex were mapped onto the structure of 5YVT as shown in Figure 7. It was observed that the regions which spanned most of the predictions by BIPSPI (Figure 7A,B) were smaller as compared to that predicted by CoRNeA (Figure 7C,D).

CoRNeA can, however, be further optimized to reduce the false-positive rates, as well as improve the true positive predictions, by increasing the training dataset. As it is evident that the environmental features play a very important role in training the classifier and that there is a correlation between the type of secondary structures and kernel matrices used to derive these environmental features, different training sub-datasets can be used to train specifically on various combinations of secondary structures to decrease the false positive prediction by Random Forest classifiers and hence increase the specificity of the overall pipeline.

## 4. Conclusions

Predicting the pairwise interacting residues for any two-given pair of proteins from only the amino acid sequence still remains a challenging problem. In this study, the newly designed pipeline CoRNeA addresses some of the challenges for predicting the PPI interfaces, such as applicability to eukaryotic PPI and high false-positive rates, by incorporating Co-evolution information and intra-contacts for improving the precision and recall of the pipeline. Overall, by including the intra-contact information through network analysis, the false positive rates reduced by 10-fold, hence improving the true negatives for the 4 examples tested with the known crystal structure in this study. Thus, this pipeline can be utilized to predict the interface residues as a pairwise entity and also to understand folding of the individual proteins though intra-contact predictions. Obtaining the structural information of proteins individually, as well as in complex with their interacting partners, is a tremendously challenging problem especially for large multimeric complexes. CoRNeA can be utilized to identify the minimal interacting regions in the heterodimers for its biochemical reconstitution, which can then be utilized in structure elucidation studies. The Random Forest output in terms of probabilities obtained from CoRNeA can also be used as a starting point for protein docking studies in cases where 3D structure models (experimental- or homology-based) are available. The web server is currently under development and the R codes, along with the trained models, are available on GitHub (https://github.com/krits04/cornea).

## Figures and Tables

**Figure 1 biomolecules-10-00938-f001:**
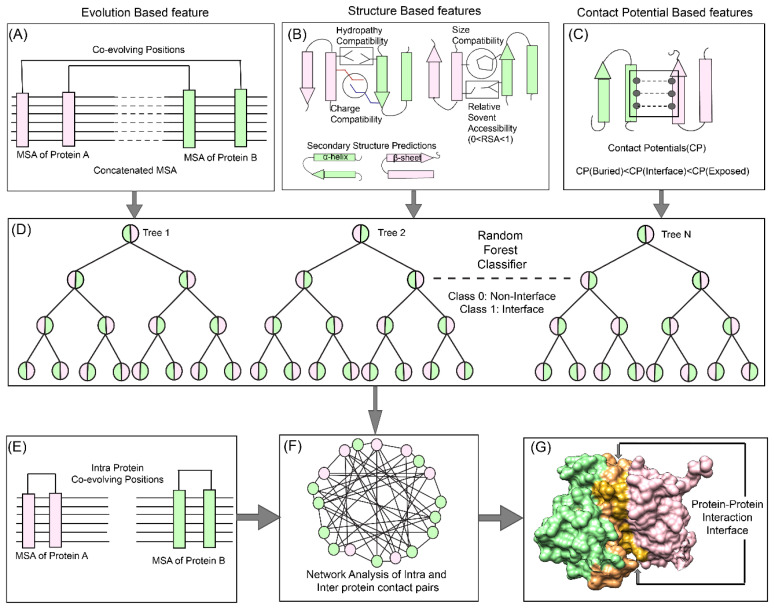
Co-evolution, Random Forest (machine learning (ML) method), and Network Analysis (CoRNeA) pipeline for predicting co-evolving contact forming residues in an interacting pair of proteins. The method for predicting the protein-protein interaction interface consists of three levels. The top panel depicts the features used for machine learning pipeline. (**A**). Evolutionary information-based (Co-evolution) (**B**) Structure-based (Charge, Size, Hydropathy, Secondary structure, and Relative solvent accessibility) and (**C**) contact potential- based features (both for buried and exposed residues). (**D**) Random Forest classification where pairwise values for both proteins are considered depicted in half green and pink circles for binary classification (Class 1: protein interface, Class 0: non-interface). The bottom panel depicts the application of network analysis by combining intra and inter-protein contact predictions for reducing the false positives. (**E**) Prediction of intra-contacts of Protein A and B. (**F**) Combined network analysis of inter and intra predicted contacts. (**G**) Interface prediction for core binding protein alpha subunit with core binding protein beta subunit (protein structure database (PDB) ID: 1H9D).

**Figure 2 biomolecules-10-00938-f002:**
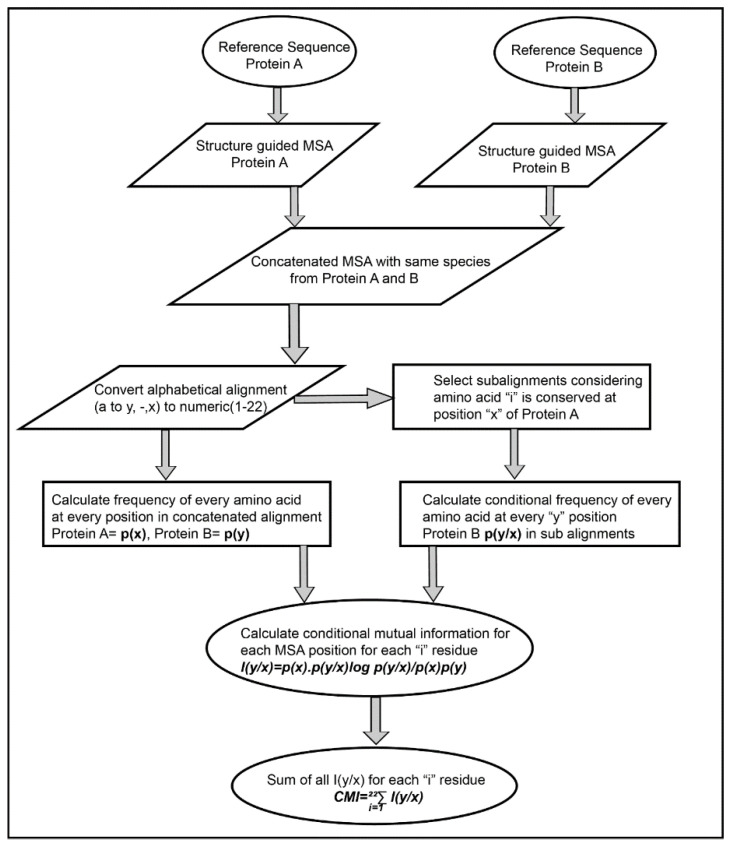
Flow chart representing an algorithm for calculating inter-protein co-evolving positions from multiple sequence alignments.

**Figure 3 biomolecules-10-00938-f003:**
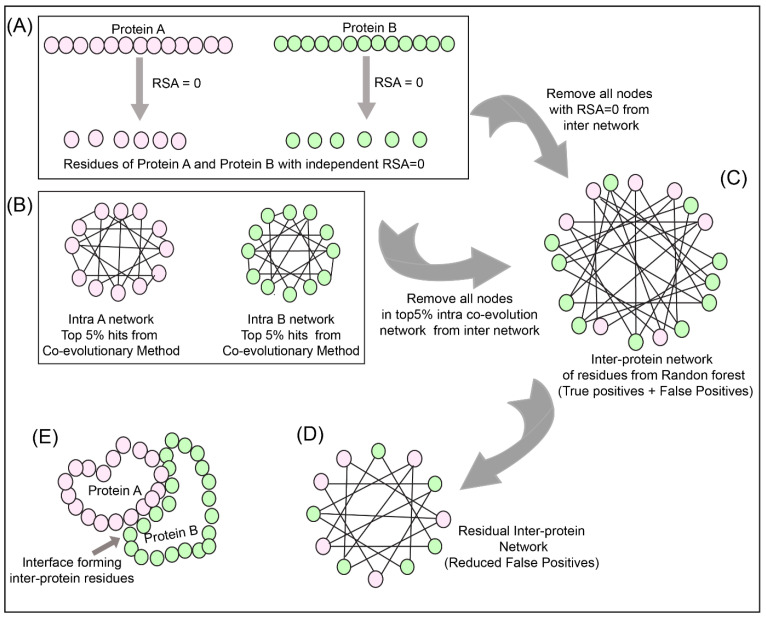
Network analysis of intra and inter-protein contacts. (**A**) Extraction of residues with relative solvent accessibility (RSA) = 0 for Protein A and B. (**B**) Intra-contact prediction for Protein A and B (top 5% co-evolving residue pairs). (**C**) Predicted inter-protein network from Random Forest classifier. (**D**) The false-positive inter-protein residue pairs obtained from the Random Forest classifier are reduced by removing nodes having RSA = 0 for Protein A and B, as well as the top 5% co-evolving intra-protein residues of Protein A and B. (**E**) Analysis of the inter-contact from residual network onto the structure of Protein A and B.

**Figure 4 biomolecules-10-00938-f004:**
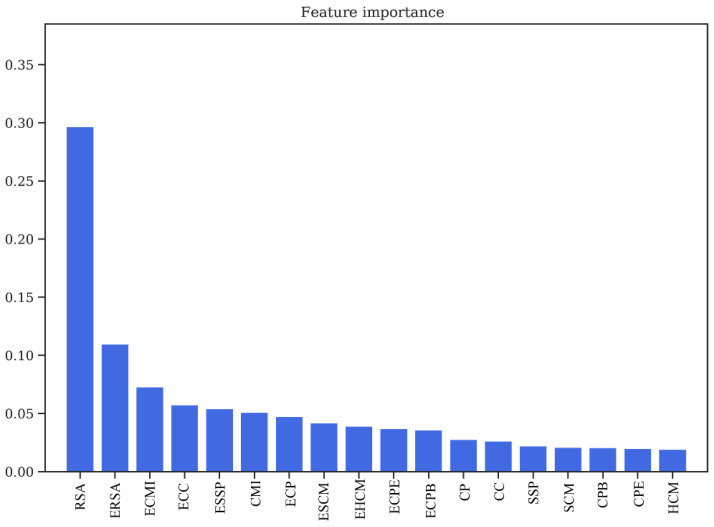
Feature importance obtained from Random Forest classifier/relative solvent accessibility (RSA/ERSA) and Co-evolution Scores (ECMI/CMI) as two of the most important features in training the model. RSA: Relative Solvent Accessibility. ERSA: Environment Relative Solvent Accessibility. ECMI: Environment Conditional Mutual Information. ECC: Environment Charge Compatibility. ESSP: Environment Secondary Structure Prediction. CMI: Conditional Mutual Information. ECP: Environment Contact Potential. ESCM: Environment Structure Compatibility Matrix. EHCM: Environment Hydropathy Compatibility Matrix. ECPE: Environment Contact Potential for Exposed residues. ECPB: Environment Contact Potential for Buried residues. CP: Contact Potential. CC: Charge Compatibility. SSP: Secondary Structure Prediction. SCM: Structure Compatibility Matrix. CPB: Contact Potential for Buried residues. CPE: Contact Potential for Exposed residues. HCM: Hydropathy Compatibility Matrix.

**Figure 5 biomolecules-10-00938-f005:**
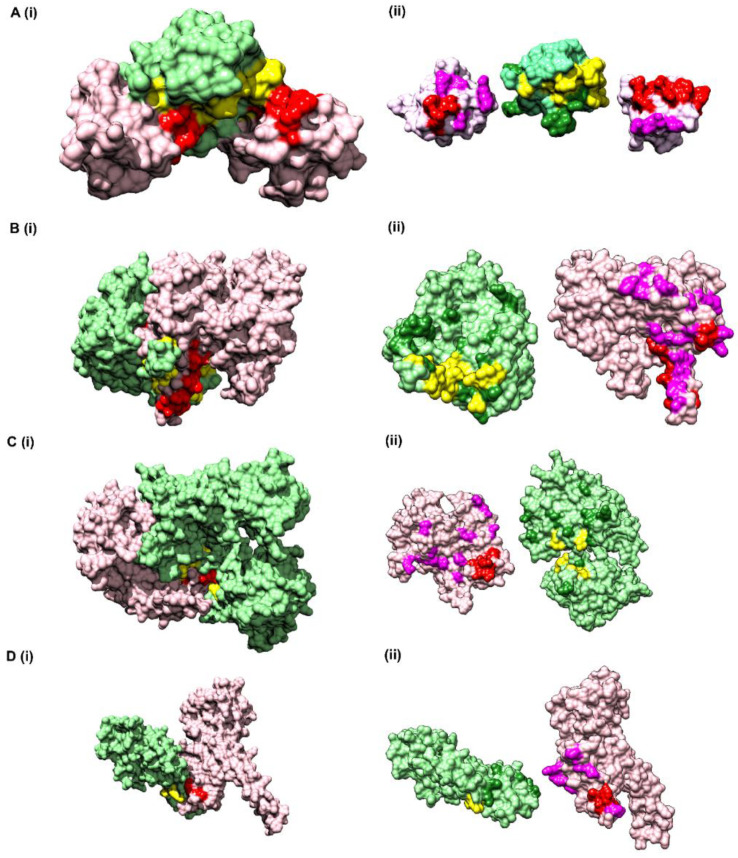
Prediction of interface hotspots on test datasets using CoRNeA. Predictions of the interface residues for 4 test datasets were mapped onto their crystal structures, (**A**) PDB ID: 1GCQ, (**B**) PDB ID: 5YVT, (**C**) PDB ID: 1Y8R, (**D**) PDB ID: 3CQC. The first column (**i**) for all four datasets depict surface representation where Protein A is colored in pink and Protein B in light green; interface residues predicted using CoRNeA for Protein A (red) and Protein B (yellow) are depicted as spheres. The second column (**ii**) depicts open book representation of the interface residues where the interface hotspots predicted by PISA and not by CoRNeA are colored as purple for Protein A and forest green for Protein B.

**Figure 6 biomolecules-10-00938-f006:**
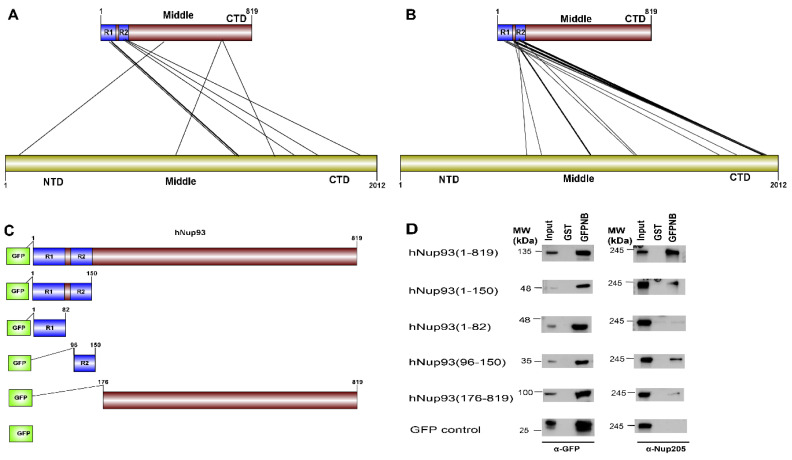
Prediction and validation of interface regions for Nup93-Nup205 (**A**) Cross-linking based mass spectrometry defined proximity regions between Nup93-Nup205 (adapted from Kosinski et. al, 2016 [51]). (**B**) Top 10% regions predicted by CoRNeA. Edges in bold depict three most significant regions (N-terminal of Nup93 with C-terminal of Nup205) (details in Appendix A). (**C**) GFP-fused deletion constructs for Nup93 for validating the predictions. (**D**) Immunoprecipitation results depicting N-terminal region (1-150) and R2 regions (96-150) of Nup93 specifically interact with endogenous Nup205. GFPNB: GST-anti-GFP-nanobody.

**Figure 7 biomolecules-10-00938-f007:**
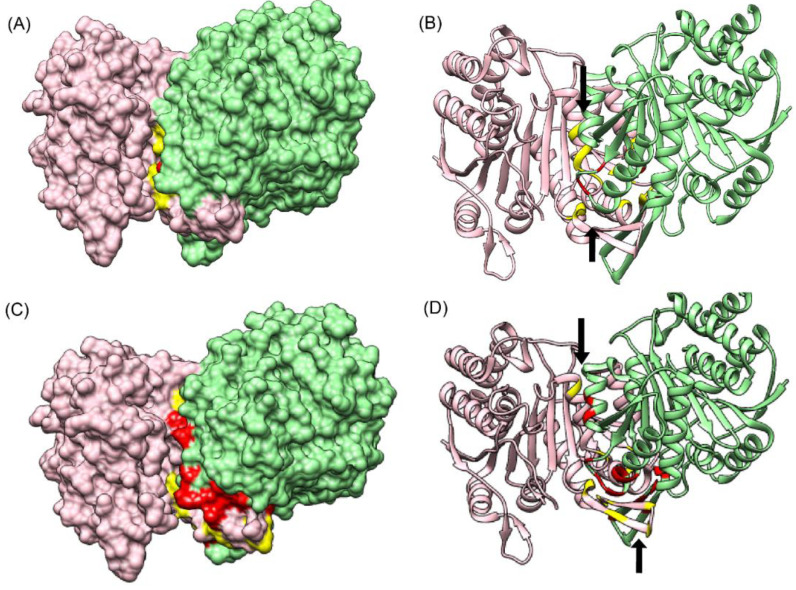
Comparison of CoRNeA with BIPSPI (**A**) Surface representation of 5YVT depicting interface residues predicted by BIPSPI. (**B**) Cartoon representation of interface residues predicted by BIPSPI. (**C**) Surface representation of 5YVT depicting interface residues predicted by CoRNeA. (**D**) Cartoon representation of interface residues predicted by CoRNeA. Chain A in pink and their respective interface residues are shown in yellow. Chain B in green and its interface residues are depicted in red. The black arrows indicate the regions of interface predicted by CoRNeA/BIPSPI.

**Table 1 biomolecules-10-00938-t001:** List of known methods for predicting Protein-Protein Interaction Interfaces (PPII).

Type	Method	Input	Output
Homology-based	PredUs	Structure	Single residue scores
	PS-HomPPI	Sequence	Pairwise residue scores
	PriSE	Structure	Single residue scores
Machine learning-based	SPPIDER	Structure	Single residue scores
	PINUP	Structure	Single residue scores
	PAIRpred	Sequence or Structure	Pairwise residue scores
	PIER	Structure	Single residue scores
	ProMate	Structure	Single residue scores
	Cons-PPISP	Structure	Single residue scores
	Meta-PPISP	Structure	Single residue scores
	CPORT	Structure	Single residue scores
	WHISCY	Structure and MSA	Single residue scores
	InterProSurf	Structure	Single residue scores
	VORFFIP	Structure	Single residue scores
	eFindSite	Sequence	Single residue scores
	BIPSPI	Structure or Sequence	Both pairwise and single residue scores
	PSIVER	Sequence	Single residue scores
	ComplexContact	Sequence	Pairwise scores
Co-evolution-based	Direct Coupling Analysis (DCA)	Sequence	Pairwise scores
	EvComplex	Sequence	Pairwise scores
	ComplexContact	Sequence	Pairwise scores

**Table 2 biomolecules-10-00938-t002:** Evaluation statistics (five-fold cross validation) for Random Forest classifier with and without environmental features.

Model	Accuracy	AUC_ROC_	Precision (Weighted Average)	Recall (Weighted Average)	F1-Score (Weighted Average)	Specificity	MCC
Without environment features	80%	0.66	0.81	0.81	0.81	0.88	0.318
With environment features	86%	0.76	0.86	0.85	0.86	0.90	0.499

**Table 3 biomolecules-10-00938-t003:** Confusion matrix statistics for all test cases before and after network analysis.

PDB ID	Method	True Negative	False Positive	False Negative	True Positive
1GCQ	Random Forest	2954	967	56	25
	After Network and Convolution	3575	317	56	42
5YVT	Random Forest	111,075	8767	100	64
	After Network and Convolution	118,880	962	134	30
1Y8R	Random Forest	213,290	7993	134	23
	After Network and Convolution	220,403	880	139	18
3CQC	Random Forest	59,214	2028	35	13
	After Network and Convolution	61,008	234	42	6

**Table 4 biomolecules-10-00938-t004:** Comparison of predictions from CoRNeA with BIPSPI and ComplexContact.

Test Dataset	Method	Expected No of Residues within 10 Å	Number of True Positives with Probability More Than 0.5	Number of False Positives with Probability More than 0.5
PDB ID: 1GCQ	BIPSPI	108	0	N/A
	ComplexContact		0	9
	CoRNeA		42	317
PDB ID: 5YVT	BIPSPI	164	24	1210
	ComplexContact		0	30
	CoRNeA		30	537
PDB ID: 1Y8R	BIPSPI	157	1	57
	ComplexContact		1	52
	CoRNeA		18	880
PDB ID: 3CQC	BIPSPI	48	0	1
	ComplexContact		0	N/A
	CoRNeA		6	240

The numbers depicted for CoRNeA are post convolution of prediction matrix. For 1GCQ the total number of expected contacts and true positives are for both chain combinations, i.e., Chain A and C; Chain B and C.

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
