# Peer review of "CoRNeA: A Pipeline to Decrypt the Inter-Protein Interfaces from Amino Acid Sequence Information"

_biomolecules, 2020, doi:10.3390/biom10060938_

Round 1

Reviewer 1 Report

A pipeline to decrypt the inter protein interfaces from amino acid sequence information

The authors present a new method, Co-evolution Random forest (ML method) and Network Analysis (CoRNeA), implemented in R for the identification of inter protein interfaces from amino acid sequences.

The introduction is very good, the authors present many methods available in the field. They also describe the problem with the focus of the current methods on prokaryotic complexes. The pipeline also is fully described, for example, Figure 1 does a good job to explain the workflow. The method is very complex, however, each step was carefully explained, with detailed descriptions and figures. Interestingly, the authors were able to validate experimentally some of the predictions. The method combines many different flows of processing information: evolution-, structure-, contact-based with random forest classifier, plus a mechanism to reduce the false positives. The data set was composed of 42 complexes with 500 residue pairs, then the imbalance problem was addressed.

Overall I like to work and appreciate the effort of the authors. I have two major concerns:

  1. At the moment, the method is compared to BIPSPI based on 4 cases. Of course, these cases could be cherry-picked. Would it be possible to compare the method to BIPSPI based on all complexes used for training (for example by re-training the method on all but this one case) and then compare it to the other method? I would like the authors to comment on that.
  2. I appreciate the Github repository. However, I think this would be super useful for a new user to add at least one full demo (step by step, the best would be to pick one of the complexes from the manuscript) how to proceed with so many single scripts put there at the moment. That I think would be also great for the tool to make a better reach.

Minor:

- I think it should be 'in silico', not 'in-silico', in the same way as the authors later use 'ab initio',

- Figure 1. I would name the complex, "Interface prediction for XXXX complex PDB ID: 1H9D." And BTW, https://www.rcsb.org/structure/1H9D is Aml1/cbf-beta/dna complex, is this correct PDB ID?

- Evaluation statistics (five-fold cross validation )for random forest classifier with and ->

- Evaluation statistics (five-fold cross validation)for random forest classifier with and (space removed)

- Four eukaryotic protein complexes with known 3D crystal structures were used to access the predictability of the complete pipeline CoRNeA. The interface residues of these protein complexes were predicted using model trained on 42 protein complexes of the affinity database. (access or assess?)

- Positives with Dataset residues within 10Å -> 10 Å (extra space)

Reviewer 2 Report

Authors present here a robust study aiming at a sequence-based prediction of residue contacts depicting protein-protein interfaces by Random Forest classifiers. The challenge is high because sequence-based predictors of PPI render high rates of false positives. Although the high rate of false positive residues is still important in the predicted PPI interface, suitable efforts have been deployed in term of methodology to reduce at best this behaviour. To my mind, it could be emphasised with a broader benchmarking evaluation with other published methods (see below). I really appreciated the further analysis of inter domain contacts in term of secondary structures given the importance of environment features observed retrospectively. For all these reasons, I recommend the publication with the following suggested minor revisions of this very valuable study in Biomolecules.

    1. Could the authors justify why they chose particularly these four PDB structure for application/evaluation of CoRNeA ?
    2. It would have been very valuable for the study to integrate other actual sequence-based predictors (eFindSite, PairPRED, PSIVER, ComplexContact) into the section 3.7 dedicated to the benchmarking evaluation of CoRNea.
    3. I strongly recommend to brighten the table 3, only displaying the confusion matrix before and after network analysis but storing the data obtained from the all 4 crystal structures. Second column and third lines are useless.
    4. In conclusion, authors support the idea that number of false positives is divided by 10 fold after network analysis to remove intra contacts. But they should also highlight the relatively low number of false negatives (around 50%).
    5. Finally it would be better to put the RF classifier output into the perspective to be used with additional meaningful data for structure elucidation studies or as starting points for protein-protein docking. For example, even after reducing significantly the number of false positive residues, we get only 10% (42/[42+317]) chances to predict a true positive residue.

Author Response

This manuscript is a resubmission of an earlier submission. The following is a list of the peer review reports and author responses from that submission.

Round 1

Reviewer 1 Report

I have the following comments.

  1. The format of the thousands separators in the manuscript is confusing, e.g., line 86 and line 212.
  2. The introduction part of the manuscript is well-prepared, with clear and sufficient background description.
  3. The “Result and Discussion” section is not easy to comprehend, however, perhaps due to that many critical data are only presented in the supplementary file, not in the main text, for example, the table S3 and S8.
  4. Other than at line 218 to 225, the authors did not say very much about the selection condition of the negative samples. How did they ensure that the negative samples are sufficiently representative so that the trained model could be reasonably generalized to other protein complexes? Another related consideration is that with only 42 protein complexes served as the training data, it is hard to believe that the trained classifier would be able to predict PPII in more general cases. The readers should have been better convinced that the classifier is not an overfitted one.

Reviewer 2 Report

In this work the authors try to decrypt the interface residues of protein-protein complexes, a central issue for improving our understanding of cellular mechanisms and diseases.

I do not think that the CoRNeA approach is very sound to this aim. According to me the main problem in this work is that the authors should verify much better the performance of their approach for example assessing it by a leave-one-out cross-validation over different benchmarks. According to me the authors should better demonstrate that CoRNeA pipeline outperforms state-of-the-art methods (PPiPP, PAIRpred, GCNN and ECLAIR). Additionally, several CAPRI targets could also tested as an independent evaluation benchmark. I do not see a real improvement compared to previously methods.In any case I deeply suggest to better clarify CoRNeA results when they assess the predictions using sensitivity (SN), specificity (SP), precision (PRE), accuracy (ACC), F1-measure (F1), Matthews correlation coefficient (MCC).

Why did they consider just one dataset, Affinity Database version 2.0? They could consider different datasets of protein complexes to train and evaluate the performance of CoRNeA predicting protein interfaces.  They can consider the Protein–Protein Docking Benchmark version 5 (Vreven et al., 2015) dataset, that contains 230 non-redundant protein complexes for which bound and unbound structures are available, or others.

Table 1, I do not see a better performance over FP assessed by a real statistical approach.

Test dataset reported on paragraph 3.6 is composed by four known 3D structures. According to me is too small. Moreover the authors do not report if the four structures of their test dataset are present in Affinity Dataset.

Minor: Figure 6 is useless because different features the authors try to show are not well displayed.

Reviewer 3 Report

The paper "CoRNeA: A pipeline to decrypt the inter protein interfaces from amino acid sequence information" presents a novel protein-protein interface prediction method. The paper is interesting and well presented but I have some major concerns that must be addressed before publication.

By reading the abstract, it is not clear if the method performs pairwise PPI prediction or single protein interface prediction. 

In the introduction, pairwise and non-pairwise protein interface prediction methods are presented together (i.e. SPPIDER and PriSE are non-pairwise interface prediction method, while PS-HomPPI is a pairwise method). The authors should clearly separate these two categories as the two problems are different. The authors should also add a table summarizing all the methods presented in the introduction.

lines 68-70: "Secondly, most of these methods have been tested on prokaryotic proteins and have a limitation of predicting only for a maximum combined length of 1500 residues per protein pair." Where did the authors get the 1500 residues figure? Please explain.

lines 74-75: "Overall these methods could not perform with similar accuracy when applied to eukaryotic complexes." Again, where do the authors get this result? Did they test this themselves? Can you present the results these methods achieve on some dataset of eukaryotic complexes?

In lines 76-84, the authors explain the differences between eukaryotic and prokaryotic proteins, and in the next paragraph they state that "it is important to develop a method specific for eukaryotic predictions". and that a dataset of eukaryotic proteins was used to train the classifier. What happens if the classifier is applied to prokaryotic proteins? How well does it perform? The claim (lines 574-575) that "the newly designed pipeline CoRNeA addresses some of the challenges for predicting the PPI interfaces such as applicability to eukaryotic PPI" feels somewhat unsupported because, for instance, the contact potentials you used (by Miyazawa and Jernigan) were not specifically extracted from eukaryotic proteins only.

The training set (42 complexes from the Affinity Database version 2.0) seems quite small to me. The test set is even smaller (4 complexes). The authors must test their method on a larger dataset. 4 complexes are not enough to provide a statistically significant result and make any conclusions on the applicability of the method. Anyone could cherry-pick 4 complexes where their method performs well. I suggest the authors to have a look at the protein-protein docking benchmark v5.0 and ProPairs (https://propairs.github.io/) datasets to increase both their training and testing sets. Moreover, the authors should ensure that there is no redundancy (and no overlap between training and test sets) in the datasets they use to guarantee the general applicability of their method. 

The reported ROC curves and the related AUC are pointless. The triangle-like shape of the ROC curves indicates that only one sensitivity/specificity point was plotted, thus the authors should just report these sensitivity/specificity values instead of the AUCs.